# Connecting energetics to dynamics in particle growth by oriented attachment using real-time observations

Lili Liu [1,5], Elias Nakouzi[1,5], Maria L. Sushko [1], Gregory K. Schenter[1], Christopher J. Mundy[1,2], Jaehun Chun [1,3,6]✉ & James J. De Yoreo[1,4,6]✉

The interplay between crystal and solvent structure, interparticle forces and ensemble particle response dynamics governs the process of crystallization by oriented attachment (OA), yet a quantitative understanding is lacking. Using ZnO as a model system, we combine in situ TEM observations of single particle and ensemble assembly dynamics with simulations of interparticle forces and responses to relate experimentally derived interparticle potentials to the underlying interactions. We show that OA is driven by forces and torques due to a combination of electrostatic ion-solvent correlations and dipolar interactions that act at separations well beyond 5 nm. Importantly, coalignment is achieved before particles reach separations at which strong attractions drive the final jump to contact. The observed barrier to attachment is negligible, while dissipative factors in the quasi-2D confinement of the TEM fluid cell lead to abnormal diffusivities with timescales for rotation much less than for translation, thus enabling OA to dominate.

[1] Physical Sciences Division, Pacific Northwest National Laboratory, Richland, WA 99352, USA. [2] Department of Chemical Engineering, University of Washington, Seattle, WA 98195, USA. [3] Benjamin Levich Institute, CUNY City College of New York, New York, NY 10031, USA. [4] Department of Materials Science and Engineering, University of Washington, Seattle, WA 98195, USA. [5]These authors contributed equally: Lili Liu, Elias Nakouzi. [6]These authors jointly supervised this work: Jaehun Chun, James J. De Yoreo. ✉email: Jaehun.Chun@pnnl.gov; James.DeYoreo@pnnl.gov

Oriented attachment (OA)[1–5], an important crystal growth mechanism, involves face specific attractive interactions between nanocrystals. These interactions facilitate the approach, capture and attachment of the crystals on interfaces with a common crystallographic orientation to produce a wide variety of crystal morphologies including rods, chains, multipods and branched nanowires[4]. To date, OA has been cited as a mechanism for crystal growth in many systems including semiconductor and metal oxides ($TiO_2$[1,6,7], $Fe_2O_3$[8,9], ZnO[10–12], CuO[13], $Sb_2O_3$[14], $MnO_2$[15], $CeO_2$[16], and $SnO_2$[17]) and oxyhydroxides (FeOOH[18] and CoOOH[19]), metal chalcogenide semiconductors (CdS[20], ZnS[21,22], and PbSe[23]), metals and metal alloys (Ag[24] and Au[25]), and carbonates ($CaCO_3$)[26].

Important physical insights into OA events have been recently obtained from real-time observations using in situ microscopy techniques. For example, previous studies used liquid phase (LP)-TEM to investigate the dynamics of OA events in the iron oxide[9], Pt-Fe[27], and Au[28,29] systems, while others employed AFM-based techniques to measure the forces between crystal surfaces, as well as their dependence on solution composition and structure for $TiO_2$[7,30,31], ZnO[12] and muscovite mica[32]. In the case of iron oxide, two nanoparticles approach each other and rotate continuously until their lattices are perfectly aligned. At that moment, the particles jump to contact and fuse into a single crystal[9]. Analysis of the particle motion showed that Coulomb forces alone could account for the jump to contact with an effective difference of only one net unit charge on the surface of the particles, but a detailed analysis of other potential forces was not carried out. By comparison, ligand-coated gold nanoparticles experience a short-range barrier due to steric repulsion before undergoing attachment[29]. In the case of gold nanorods, the potential energy as a function of angle and distance was derived for that system, but the underlying force components (e.g., van der Waals, dipole–dipole,

etc.) were not examined[28]. Thus, despite direct evidence of OA, the nature and scaling of the underlying forces is not understood and hence a comprehensive understanding of the mechanism remains lacking. In particular, a quantitative understanding of the relationship between the structure and surface chemistry of the nanocrystals, the structure of the solvent between crystal surfaces —including the distribution of electrolyte species—the forces that attract or repel the crystals, and the resulting particle response dynamics has not been established. Consequently, although OA plays an important role in controlling particle morphology and composition, the rational design of complex materials using OA remains a tremendous challenge. Here we take a key step towards establishing the relationships between structure, forces and response dynamics needed to address that challenge by investigating OA of the important oxide semiconductor ZnO.

ZnO readily forms quasi-one-dimensional nanostructures and is a canonical example of a functional material proposed to form via OA. Moreover, ZnO nanostructures have a broad range of applications in electronics, photonics and piezoelectric devices, and exhibit excellent chemical and thermal stability[10,11,33,34]. Fig. 1 shows ex situ TEM images of ZnO nanoparticles that have been placed in a methanol solution of 1 mM zinc ions and maintained at room temperature for a period of three months. Within this time, the nanoparticles formed elongated nanorods that extend along the [001] direction (Fig. 1a–c, Supplementary Fig. 1). High resolution imaging shows that the nanorods are single crystals, but consist of smaller, co-aligned domains. Similar results have been previously reported for several metal oxide and oxyhydroxide crystals of different geometries, and, on the basis of such data, the crystal growth mechanism has been ascribed to particle attachment[1,11]. However, as with the ex situ TEM data on nearly all systems listed above, such results do not present direct evidence for OA and provide no information about the process of

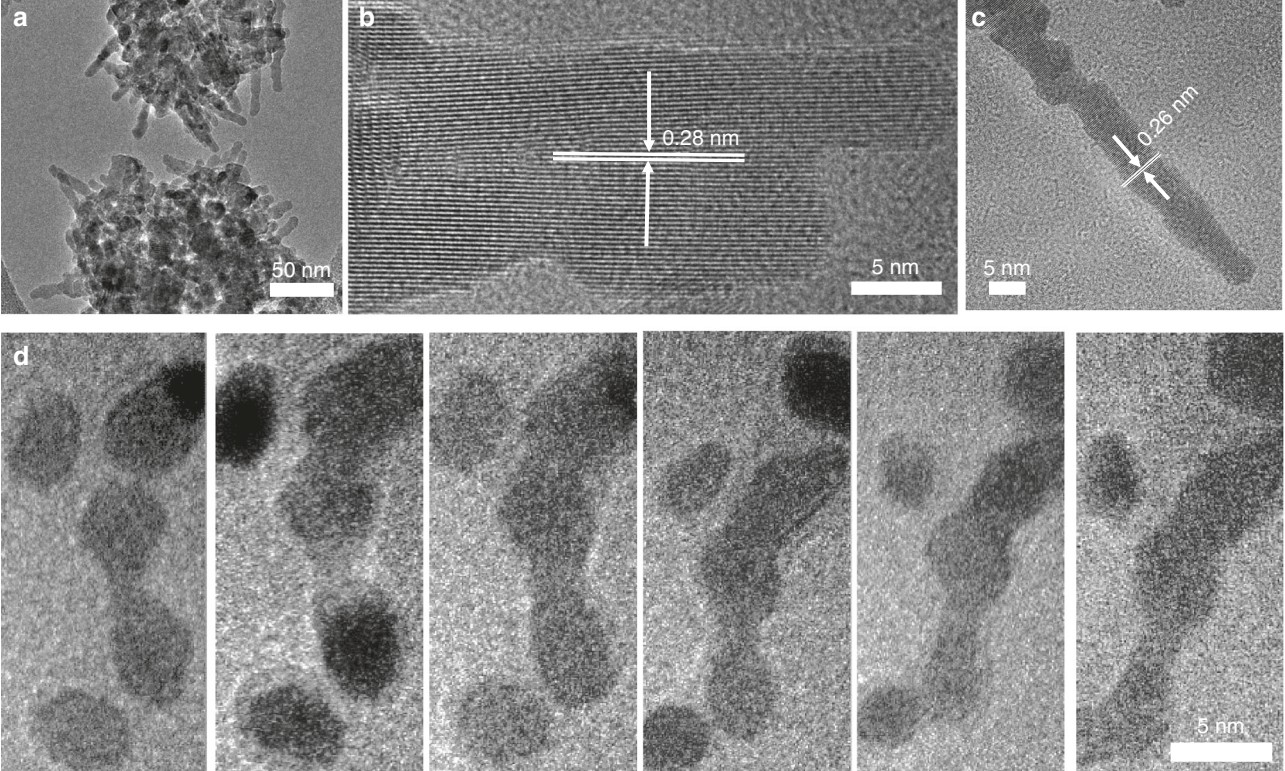

**Fig. 1 Zinc oxide nanostructures grown from solution. a–c** Ex situ TEM images of ZnO grown in methanol solution containing 1 mM $Zn^{2+}$. **d** In situ LP-TEM images of growth of ZnO chain structure in methanol solution containing 1 mM $Zn^{2+}$. Scale bar represents 5 nm.

assembly[10,11,33]. Consequently, little is known about either the dynamics or the underlying controls on ZnO growth by particle attachment.

While, in principle, LP-TEM can help to provide an understanding of growth of ZnO from precursor particles, as it has for some other systems[9,27–29,35–37], there are a number of limitations that must be overcome. First, LP-TEM observations of metal oxide growth are scarce, because most traditional synthesis methods involve hydrothermal conditions incompatible with LP-TEM. Second, e-beam induced dissolution of metal oxide nanoparticles in LP-TEM conditions is a common problem, which hampers imaging of particle assembly in systems like ZnO. Third, high resolution TEM techniques typically image processes occurring on two-dimensional (2D) surfaces instead of a bulk solution phase due to the intrinsic confinement associated with the underlying TEM membrane. We present here an experimental approach that mitigates these experimental obstacles and provide detailed analyses of the results within the framework of quasi-2D aggregation kinetics and hydrodynamics. These measurements and analyses reveal both forces and torques acting at edge-to-edge particle separations of up to 10 nm. We then employ simulations of interparticle interactions, as well as Langevin dynamics, to understand the underlying forces responsible for nanoparticle approach, coalignment, and attachment.

## Results

**Establishing conditions of ZnO stability in LP-TEM.** To overcome the need for hydrothermal conditions, we developed a facile synthesis method in which a dispersion of ZnO particles is made from methanol solutions of zinc acetate dihydrate and KOH at 60 °C, and then allowed to grow in methanol at room temperature (see "Methods" section for details). However, when these ZnO nanoparticles were imaged in pure methanol solution by LP-TEM, the particles began to dissolve almost immediately with some crystals disappearing completely within 30 s (Supplementary Fig. 2a). Similar experiments in water led to even more rapid dissolution. To mitigate this problem, we placed the nanoparticles in methanol solutions with additional zinc acetate. While at $[Zn^{2+}] = 0.01$ mM, dissolution still occurred (Supplementary Fig. 2b), when $[Zn^{2+}]$ was increased to 1 mM dissolution was no longer observed (Supplementary Fig. 2c). (See the Supplementary Discussion for a rationale for ZnO dissolution in water and pure methanol under the action of the electron beam).

By mitigating the dissolution problem and optimizing the imaging conditions, we were able to monitor the behavior of the ZnO nanoparticles over long timescales. Representative in situ TEM images in Fig. 1d clearly demonstrate that particle attachment is a viable growth pathway in this system. Specifically, at $t = 0.5$ s, two particles merged to form a dimer. A third particle then contacted one end of this dimer to form a trimer at $t = 40.5$ s (Supplementary Movie 1). Additional end-to-end attachments generated a nanoparticle chain, confirming that individual nanoparticles can be the fundamental building blocks for the formation of ZnO nanorods seen in Fig. 1a–c.

**Evidence for oriented attachment.** To delineate whether two approaching NPs align their crystallographic axes prior to attaching or simply attach while misaligned and subsequently coarsen into a single crystal, we monitored the individual trajectories and attachment events of a number of ZnO nanoparticle pairs. For some particles, it was possible to identify their relative orientations ($\theta_r$) based on their (002) lattice fringes, although they occasionally fluctuated slightly off-axis. In the early stages, at an edge-to-edge separation $h = 4.5$ nm, the particles exhibited an angle $\theta_r = 43°$ between their respective (002) faces (Fig. 2a–f,

Supplementary Movie 2). As the particles came into closer proximity, they also rotated, and $\theta_r$ progressively decreased to 15° at $t = 54$ s ($h = 3.1$ nm), and then to 4° at $t = 63$ s ($h = 2.6$ nm). As the two particles aligned to a perfect lattice match, OA was accomplished via a jump-to-contact across a gap of 0.96 nm (~5 nm center-to-center spacing). Attachment thus occurred along the [001] crystallographic axis, leading to the formation of a larger particle (Fig. 2f) for which the single-crystal nature was verified by the corresponding Fourier transform (Inset, Fig. 2f).

Fourier transform analyses confirmed this picture of attachment. The intensities from the power spectra summed along the orientation angle ($\theta$) and the corresponding intensity profiles plotted versus time (Fig. 2g) for two particles both before and after coalescence show that the two particles rotated and aligned their crystallographic axes within 20 s prior to contact. After attachment, the resulting particle maintained a similar orientation and its rotational dynamics were slower. In all three cases for which it was possible to measure relative particle orientations just prior to attachment, rotation to co-alignments was clearly observed (Supplementary Figs. 3, 4, Supplementary Movie 2–3).

**Dynamics of particle attachment.** We further examined a sequence of images to determine the dynamics of particle approach and shape evolution by analyzing a number of attachment and coalescence events (Fig. 3a–c; Supplementary Movie 4). The particles highlighted with similar colors in Fig. 3a eventually merged into a single crystal during the observation time. In each case, the two particles steadily moved towards each other with some minor fluctuations in their trajectories, followed by a jump to contact. After contact, the connecting neck between the particles rapidly vanished (Fig. 3d). To evaluate the contribution to particle growth from the attachment versus ion addition, we derived the expected dependence of projected particle area on volume for the case where the volume of the final particle is simply given by the sum of the volume of the two primary particles. The evolution of the resulting normalized particle area defined as $1/(A_1^{3/2} + A_2^{3/2})^{2/3}$ (Fig. 3a–c; derived formula shown in Supplementary Discussion) between attaching particles (Fig. 3e) shows that the area of the newly merged particle consisted approximately of the sum of the areas of the two individual particles. Thus, the elimination of the neck occurred through rapid diffusion of surface atoms into the concave regions rather than by ion addition from solution. In fact, no particle growth by ion addition from solution was observed anywhere in the imaging area during the observation time (Supplementary Fig. 5), showing that particle attachment was the predominant pathway of particle growth in our system. Plots of the edge-to-edge particle separation ($h$) versus time (Fig. 3f) show that the particles slowly but steadily approached one another before jumping to contact at $h = 1.08$ ($\pm 0.33$) nm, which is comparable to distances previously reported for Au and ferrihydrite nanoparticles[9,38].

To determine the underlying free energy landscape across which particles diffuse and interact, we analyzed the collective behavior of particle ensembles (Fig. 3a–c; Supplementary Movie 4) by calculating the radial distribution probability function (RDF) (Fig. 4a) according to $g(h) = \frac{1}{2\pi h \Delta h} \frac{N_h}{\rho}$, where $N_h$ is the number of particles within $[h - \Delta h/2, h + \Delta h/2]$ and $\rho$ is the number of particles per unit area[39]. For this analysis, all the particles in the imaging area were considered and care was taken to account for the truncated sampling areas for particles near the frame edges[28,40]. A frame-by-frame analysis shows that the RDF did not change appreciably during the course of the experiment, and hence provides representative information about the quasi-steady state distribution of particle separations (Supplementary Fig. 6). Moreover, the constancy of the RDF implies that particle

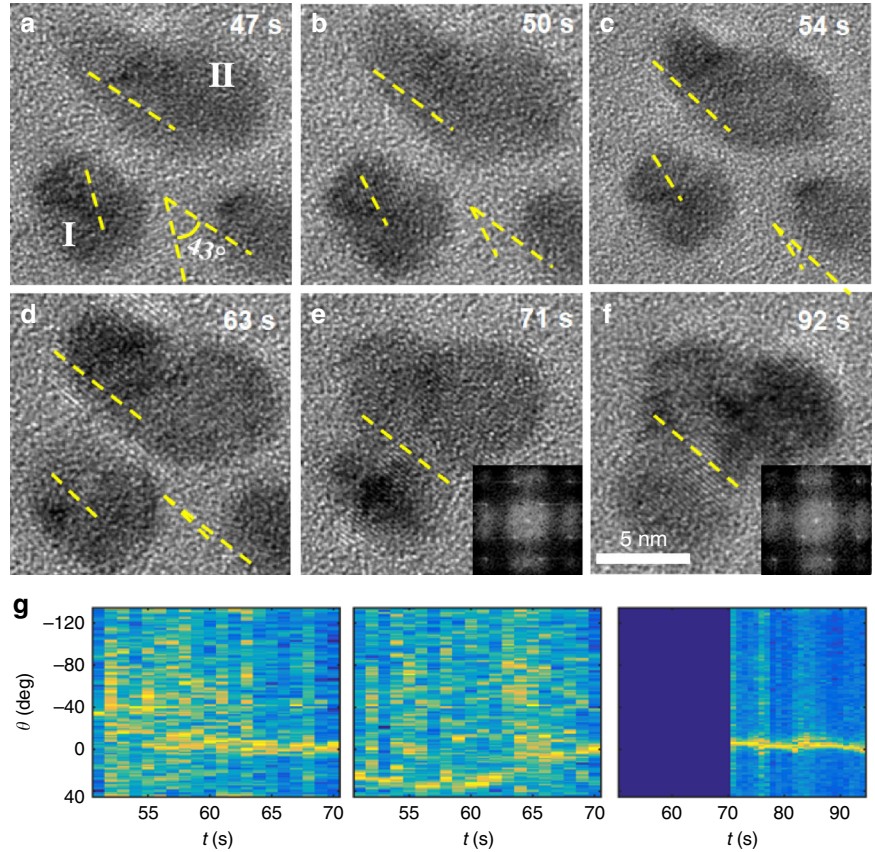

**Fig. 2 Oriented attachment of ZnO nanoparticles. a–f** Sequence of LP-TEM images showing OA of ZnO nanoparticles in methanol solution containing 1 mM $Zn^{2+}$. Scale bar is 5 nm. **g** 1D intensity profiles versus time created from FFT analyses (particle I: left panel, particle II: middle, fused particle: right) of image sequence show that the two particles rotated and aligned their crystallographic axes within 20 s prior to contact. This result is manifest as a trace of high contrast (yellow) in the panels of **g**.

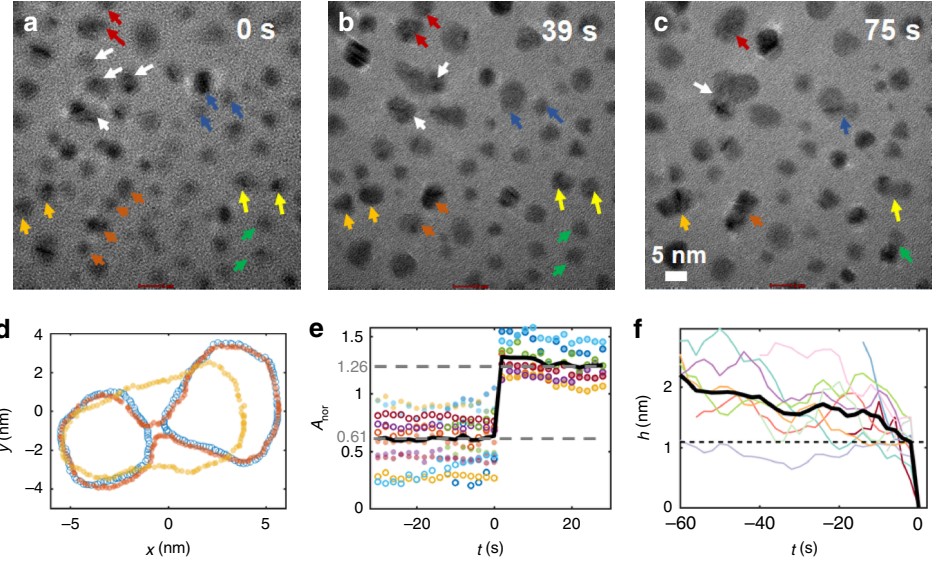

**Fig. 3 LP-TEM data of ZnO nanoparticle diffusion and coalescence in methanol containing 1 mM $Zn^{2+}$. a–c** Time sequence of images, particles marked by the same colors eventually coalesce. **d** Particle contour of two merging particles. Blue to red to yellow markers denote early to late frames. **e** Normalized particle area versus time during the attachment process. Filled and open markers denote individual particles in a particle pair while colors designate distinct particle pairs. The gray dotted lines denote the average particle size prior to and after attachment. **f** Edge-to-edge separation of pairs of approaching particles showing jump to contact. Colors designate distinct particle pairs.

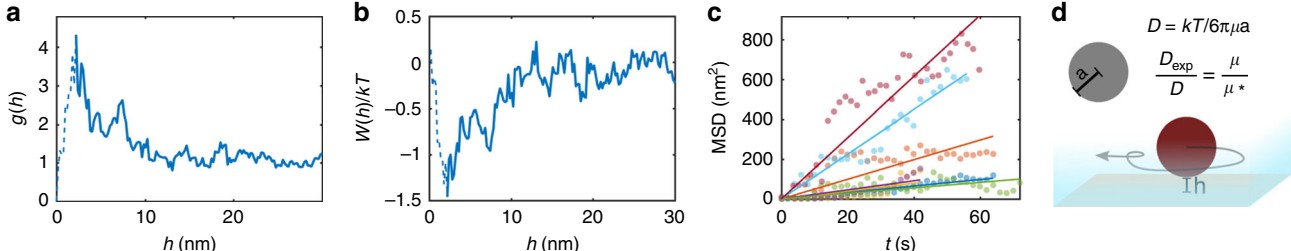

**Fig. 4 Analysis of nanoparticle interactions and diffusion. a** Radial distribution function, **b** $W(h)/kT = -\ln(g(h))$. Note that the decrease in $g(h)$ at small $h$ values is not due to a short-range repulsion, but rather to the particle attachment events that rapidly remove any particles from this region, as well as the inability to sample probability at $h = 0$, **c** mean squared displacement of diffusing particles. Color markers correspond to different example particles. **d** Scheme depicting significantly reduced particle diffusivity near the TEM membrane surface.

concentrations are sufficiently low that steric hindrance and repulsive forces due to neighboring particles are negligible. By inverting the RDF, we obtain a weak attractive potential W(h) between the particles (Fig. 4b) over relatively long separations of several nanometers. Importantly, within the uncertainties of the analysis, an attractive interaction is already apparent at 10 nm separation and no appreciable energy barrier (<1kT) is detected, which suggests a diffusion-limited process for particle attachment[29].

To validate this result, the initial particle-particle attachment rate coefficient ($k_{11}$) was calculated by assuming second order kinetics according $-dn/dt = k_{11} n^2$ under a quasi-2D approximation. Here we consider tangential motions of the particle to dominate due to the much stronger hydrodynamic lubrication forces in the normal direction[41], which originate from the confined geometry close to the TEM membrane. Therefore, $n$ is the number of monomers in the imaging area at a given time in contrast to a conventional volumetric number density in 3D cases. Analysis of the kinetic data gives $k_{11} = 1.53$ nm$^2$ s$^{-1}$ (Supplementary Fig. 7). We then calculated mean squared displacements for multiple particles (Fig. 4c) and obtained an average diffusion coefficient of 2.99 nm$^2$/s for particles of radius $a = 2.2$ nm, which is the average particle radius in the experiments (see Supplementary Discussion for details); approximately six orders of magnitude lower than expected from the Stokes–Einstein equation, but similar to the anomalous diffusivities reported for other nanoparticle systems in LP-TEM ascribed to the interactions between the particles and the membrane[42–44]. Using these values, the presumed diffusion-limited hypothesis was evaluated based on the Arrhenius equation ($k_{11}$ = collision frequency × $e^{-E/kT}$, see Supplementary Discussion for details), giving a negligible barrier for particle attachment of ~ 3kT, comparable to thermal energy and qualitatively consistent with the result derived from the RDF.

To determine the extent to which the experimentally observed coagulation rate compares to the ideal rate for non-interacting particles, we derived the stability ratio ($W_{st}$) which equals unity in the ideal case and increases as the strength of the repulsive interparticle interaction increases[45]. For the case of particles diffusing in quasi-2D, the ideal coagulation rate ($k_0$) is expressed as:

$$k_0 = \frac{2kT}{3\mu a \, G(a,h)} \frac{1}{\ln(R/2a)} \tag{1}$$

where $G(a,h)$ and $\mu$ represent the hydrodynamic resistivity and effective viscosity near the interface, respectively (see Supplementary Discussion for details). We calculate $k_0$ as 73.5 nm$^2$ s$^{-1}$, which results in $W_{st} = k_0/k_{11} = 48.0$. Since the stability ratio can span as many as six orders of magnitude for a single colloidal system, depending on the surface and solution chemistry[45], this low-to-intermediate value further supports the conclusion of a

negligible energy barrier for particle attachment and thus a diffusion-limited process. This analysis also shows that the effect of the membrane is simply to impose a high viscosity that resists particle motion, thus rendering it mass transport limited. As a consequence, the extracted radial distribution function represents a quasi-equilibrium distribution.

In addition to quantifying the attachment dynamics, these kinetic data and associated analyses have implications for estimating the probability of oriented versus non-oriented attachment. First, there is no evidence for two qualitatively distinct pathways in the kinetics data; there is clearly just one pathway with a negligible barrier (Fig. 4b). Moreover, we observed OA in three out of the three events for which it was possible to resolve particle orientations, despite the fact there are significantly less rotational configurations associated with particle alignment than there are for misalignment. In other words, if the two pathways were equivalent, non-OA should be observed far more frequently, which is not the case. Accordingly, we infer that OA is the statistically dominant pathway for this system.

This outcome might well be expected based on the analysis of particle orientation vs separation and the timescales for rotation vs translation. For the particle pair in which the rotation is resolved throughout the alignment process, rotation towards alignment begins already at 4.5 nm edge-to-edge separation. This demonstrates the existence of torques at such distances, which can rotate the particles into alignment provided the time available is sufficient. The generalized relative velocity of a particle pair is given by $U = -D \cdot \nabla(W(h)/kT)$, where $D$ is a relative particle diffusivity related to the particle hydrodynamic mobility ($M$) by $D = kTM$[46]. For our system, the abnormally slow particle diffusivity and quasi-2D nature originating from the solution structure near the TEM membrane cause significant deviation from particle diffusion in bulk solutions. Using a scaling analysis and considering the effective viscosity of the fluid in close proximity to the underlying TEM membrane, we determine that the timescale for particle rotation (~ 0.1 s) is significantly faster than that for translational motion (~60 s, see Supplementary Discussion for details). Accordingly, under the action of the experimentally documented torque, two approaching particles have sufficient time to sample all possible relative angles and accomplish a specific orientation for OA.

**The interactions that drive ZnO oriented attachment.** Our experimental results provide important comparisons with recent investigations of the forces between ZnO crystal surfaces. For example, Zhang et al. determined that the adhesive force between two ZnO nanoparticles depends on their relative orientation and is a factor of two greater for aligned particles than for those maximally misaligned, implying a preference for coaligned attachment along (0001), consistent with the above conclusions[12].

However, classical molecular dynamics (MD) simulations in that study[12], as well as that of Shen et al[47] for the case of ZnO($10\bar{1}0$) surfaces predicted that hydration forces between two approaching zinc oxide particles produce significant repulsive barriers at sub-nanometer particle separation, which contrasts with the above experimental findings that the measured energy barrier is practically negligible. Moreover, the MD simulations do not predict significant interparticle interactions beyond 1 nm, while the experimental data demonstrate torques beyond 5 nm and attractive interactions out to 10 nm. The MD results also indicate that a possible reason for the discrepancy in the barriers is that the repulsive hydration force depends strongly on the surface chemistry[47], as well as curvature in the case of small particles[48]. Importantly, molecular simulations under the low salt concentration solution conditions relevant to this study would require prohibitively large and long simulations to provide insight into the long range solution-mediated interactions and standard molecular potentials are not parametrized to capture long-range dispersion interactions between nanoparticles[32]. Thus, to understand the interactions responsible for both the negligible barrier below 1 nm and the attractive interactions and torques beyond 5 nm, it is necessary to employ theoretical frameworks capable of probing both short- and long-range interactions that are mediated by the solution. Complexities in the experimental parameters, namely the extremely small separations, reduced dielectric contrast due to methanol, and the mixed valency of the ions comprising the electrolyte, complicate extraction of specific molecular information from molecular simulations and input to simple dielectric theories for calculating particle interactions[49–51]. Consequently, we chose to consider the granularity of solvent and ions in the context of classical density functional theory (cDFT) which can access both limits[52].

A key feature of cDFT is that it captures important fluctuations beyond those described by simple dielectric theories arising from electrostatic correlations (i.e., electrostatic forces associated with spatial fluctuations of ion distributions), which are important for characterizing the solution-mediated contributions to assembly. In addition, cDFT provides a rigorous separation between packing and electrostatic contributions to driving forces of nanoparticle assembly. In the cDFT framework (Fig. 5a), inter-particle forces are dominated by electrolyte-mediated interactions through inhomogeneous charge distributions that give rise to an electrical double layer. Non-packing and non-electrostatic contributions, such as those captured in non-retarded Lifshitz theory with an isotropic dielectric tensor to describe the dispersion force, can be post-processed within this framework (see Supplementary Discussion for details).

We used the cDFT simulations to calculate forces (per area) between two ZnO flat crystal faces in 1 mM zinc acetate dihydrate solution in methanol for edge-to-edge separations of 0.5 nm ≤ $h$ ≤ 8 nm (Fig. 5a). (Limitations of the model at $h$ < 0.75 nm prevent an accurate evaluation of face specificity of the final stages of the particle desolvation and reactive molecular simulations would be more appropriate in this regime.)[53] We considered interactions between pairs of (0001)–(0001), (10-10)–(10-10), (0001)–(10-10) and (0001)–(000-1) faces. The latter have opposite charge due to the intrinsic dipole moment of the ZnO crystal lattice, which, in vacuum, leads to expression of both Zn-terminated and O-terminated faces. In order to connect to previous work that investigated the forces between oppositely charged surfaces using primitive Monte Carlo and cDFT models that treat solvent as an isotropic dielectric medium[54–56], we started by simulating the interactions between pairs of (0001)–(0001) and (0001)–(000-1) faces with surface charge patterns representative of the faces in a mean-field solvent. The full model is based on the primitive model with the addition of neutral hard-spheres representing the methanol solvent and a non-electrostatic interaction between the ions and the neutral solvent that are derived to fit hydration enthalpies of $Zn(Ac)_2$ in methanol.

The primitive model predicts no barrier to attachment for the (0001)–(0001) faces and a small barrier of <0.1 kT nm$^{-3}$ for the (0001)–(000-1) faces (Fig. 5b, c). The existence of a repulsive interaction between oppositely charged surfaces, though counter-intuitive, is expected as a consequence of ion correlation and is in agreement with Monte Carlo simulations of similar systems[54–56], where the electrostatic interactions between trivalent cations in water present in those simulations are replaced here by $Zn^{2+}$–$Zn^{2+}$ interactions in methanol, which are of similar strength due to similar Bjerrum length, $q^2/\varepsilon kT$, where $q$ is the proton charge of the ion and $\varepsilon$ is the dielectric permittivity of the medium. These

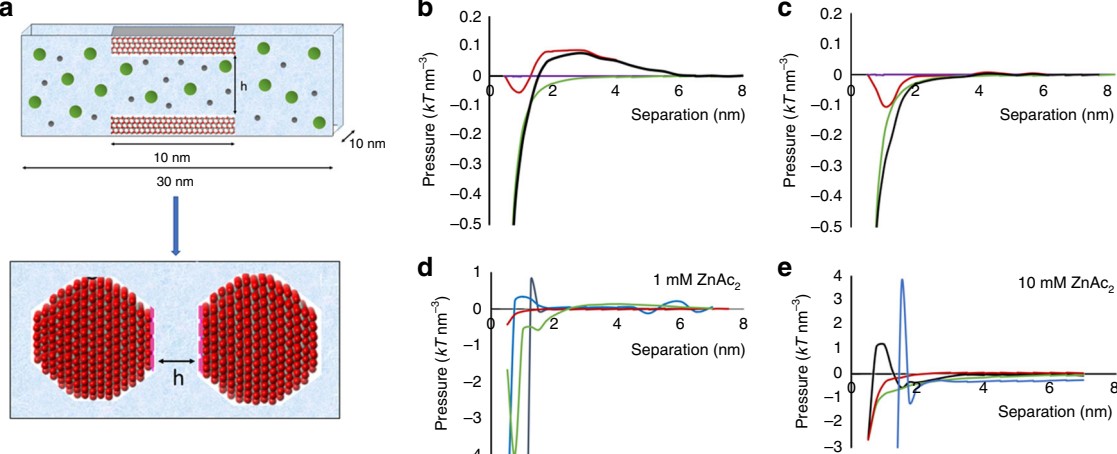

**Fig. 5 cDFT simulations of ZnO particle attachment. a** Schematic of cDFT simulation, **b, c** Disjoining pressure vs edge-to-edge separation between **b** two oppositely charged (0001)–(000-1) faces and **c** like-charged (0001) faces with aligned patterns of charges in 1 mM zinc acetate dihydrate solution in methanol calculated using the primitive model showing electric double layer (blue), ion correlation (red), hard sphere (purple), and van der Waals (green), and total (black) components of pressure. **d, e** Total disjoining pressure between pairs of ZnO facets in **d** 1 mM and **e** 10 mM zinc acetate dihydrate solution in methanol calculated using the full model for (0001–0001) (black), (10-10)–(10-10) (green), (0001–10-10) (red), and (0001–000-1) (blue) configurations. The charge patterns for (0001)–(000-1) faces are misaligned by 20 degrees (the corresponding data for aligned pattern of charges are provided in Supplementary Fig. 8).

results show that, even in the primitive model, electrostatic and van der Waals interactions produces negligible barriers to attachment.

The full model predicts barriers similar to those of the primitive model in that they are small or non-existent; a small salt-dependent barrier of order $kT\,nm^{-3}$ exists between (0001)–(0001) and (0001)–(000-1) faces, while the forces are almost monotonically attractive between (10-10)–(10-10) and (0001)–(10-10) faces (Fig. 5d, e). However, the repulsive barriers are now restricted to the region between ~1.0 and 1.5 nm of separation and the steep attractive region beyond the barrier begins at ~1.0 nm, consistent with the distance at which the jump to contact is observed in the experiments. Further calculations that account for the effect of appreciable curvature of the particles were performed based on a surface element integration scheme[57]. With $a = 2$ nm representing the average particle radius, the barrier to attachment between two ZnO nanoparticles with oppositely charged (0001)–(000-1) faces remains at $h \sim 1$ nm and is ~kT (Supplementary Fig. 9) in height, consistent with the experiments.

Although the prediction of the barrier is consistent between the full model and experiment, a careful interpretation of the full model is warranted. The structural aspects of the full model produce nonintuitive oscillations of the neutral hard-sphere fluid and do not appear to yield the desired homogenous state far from the interfaces (Supplementary Figs. 8, 9, and 10). The predicted double-layer also exhibits unexpected behavior in which the $Zn^{2+}$ concentration tracks with that of the counterions, an effect seen in trivalent salts at only much higher concentrations[49]. In addition, for the oppositely charged (0001)–(000-1) surfaces, the unusual oscillations in the disjoining pressure at large particle separations (>5 nm edge-to-edge) are exceedingly sensitive to an exact alignment of the surface charge patterns; the significant oscillatory ion correlation interactions between these faces occur only near exact alignment of charge patterns on oppositely charged (0001) surfaces (Supplementary Fig. 8). Therefore, while further investigation is needed to appreciate the meaning and source of these features, the full model suggests that interesting phenomena can occur due to the additional physics of desolvation in the vicinity of the nanoparticle interface. These phenomena are related to charge-regulation that may possibly play a significant role in the assembly[58].

As stated above, full cDFT framework is general enough to include explicit hard-sphere solvent and non-electrostatic mean field solvent-ion interactions that produce a more complete model, which better mimics the experimental system than either the primitive model or previous MD simulations (see Supplementary Discussion for details). However, this complexity renders the cDFT model closer to a full molecular simulation that is difficult to validate—either with more traditional molecular simulations or by performing a self-consistent check of the theory through a full simulation of the charged hard-sphere system in a hard-sphere solvent interacting via a patterned dielectric nanoparticle[59,60]. Moreover, whether a neutral hard-sphere is a meaningful representation of an associating fluid (like methanol) is an open question[61] as is the form of the non-electrostatic solvent interaction. Whether refinement of these aspects of the model will alter the predictions of oscillations in the potential at long range remains to be seen.

Previous analyses of potentials between ZnO faces based on MD simulations predicted multiple barriers far in excess of kT within about 1 nm of the surface due to the formation of hydration layers near the crystal in aqueous solution[12,47]. The inclusion of electrostatic, ion correlation and van der Waals forces appears to resolve the disagreement between the MD predictions and the experimental observation that barriers are

insignificant. However, the cDFT results highlight a critical aspect of OA in the ZnO system: none of the attachment configurations (OA or non-OA) are prohibited by energetic barriers much above kT and, in fact, the largest barriers are predicted for attachment on the crystallographically correct (0001)–(000-1) faces. Thus, the short-range barriers seen in cDFT cannot be responsible for face selectivity. This conclusion provides a rationale for the strikingly different behavior observed here than in other systems, such as iron oxide[9], in which particles diffuse freely to within close range and then sample a wide range of potential alignments until a perfect lattice matching orientation is achieved, at which point there is a sudden jump to contact. Instead, the ZnO particles observed here are attracted to one another beginning at ~10 nm separation (Fig. 4b) diffuse slowly towards one another and rotate into alignment starting at about 5 nm (Fig. 2a–c) so that they are fully aligned by the time they reach ~1 nm separation. Thus, we conclude that ZnO OA along [0001] dominates because long range torques pre-align the particles before they reach separations at which jump to contact can occur.

The source of the long-range attractive interaction and aligning torque can be understood when the inherent electric dipole of ZnO is taken into account. While the cDFT model presented above incorporates the electrostatic, van der Waals, solvation and ion correlation forces, it does not include dipole–dipole forces, which must exist, given that ZnO is a polar material presenting Zn and O terminations, respectively, at the [0001] and [000-1] faces, leading to a spontaneous dipole moment[62]. Because dipoles naturally align through both an attractive interaction and a torque, each of which scales as $1/h^3$, we explored this possibility by simulating the translational and rotational Langevin dynamics of two 2.2 nm radius spherical dipoles experiencing random Brownian forces and torques in an orientation-dependent dipole–dipole interaction potential (See Supplementary Discussion for details). We further assumed that the dipole–dipole interaction was the sole contribution to $W(h)$. The dipole moment of ZnO nanoparticles ($\mu = 779$ D) was estimated from fitting the measured $W(h)$ beyond 10 nm to $-2\,\mu^2/\varepsilon r^3$ (i.e., asymptotic behavior of the dipole–dipole interaction at larger separations) based on the aligned dipoles, where $\varepsilon = 32$ (methanol) and $r = h + 2a$ (See Supplementary Discussion for details). Note that estimates based, instead, on a previous study[10] give a value of ~ 670D, which is similar to our estimate based on fitting $W(h)$ at large $h$.

Figure 6a shows the resulting potential energy vs edge-to-edge particle separation and Fig. 6b shows the tendency towards coalignment, described by $\cos\theta_1\cos\theta_2$, where $\theta_1$ and $\theta_2$ are the orientation angles of the particle 1 and 2 with respect to a line between the particle centers, $r_{12}$, as a function of particle separation, based on over 10,000 ensemble trajectories starting from $h = 20$ nm (Supplementary Fig. 11, Supplementary Movie 5). A value for $\cos\theta_1\cos\theta_2$ of unity is achieved when particles are aligned "head-to-tail" ($\theta_1 = \theta_2 = 0$ or $\theta_1 = \theta_2 = 180°$). These simple dynamic calculations illustrate that both attraction and torque at large separations should be expected due to the inherent dipole–dipole interactions between ZnO nanoparticles. A further in situ TEM study with various nanocrystals having different dipole moments could test this postulation; a variation of the dipole moment would provide noticeable difference in length scale associated with both the attractive interaction and aligning torque[63].

## Discussion
The findings reported above show that ZnO nanoparticles of approximately 5 nm diameter are attracted and experience an aligning torque starting at edge-to-edge separations of more than

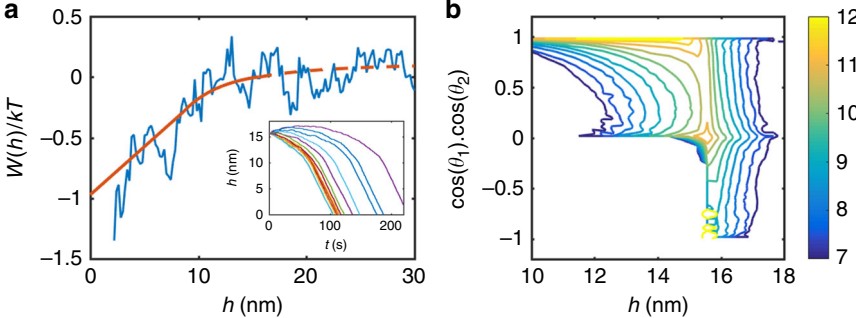

**Fig. 6 Impact of dipole–dipole interactions on ZnO particle diffusion. a** Experimental $W(r)$ and corresponding two-piece fit to a hyperbolic function at longer separations and linear function at shorter separations. Inset shows trajectories of individual particles. **b** The tendency for orientation as a function of edge-to-edge separation between two dipole particles over 10,000 ensembles with respect to the initial angles. Note that $\cos\theta_1\cos\theta_2 = 1$ corresponds to the perfect "head-to-tail" attractive parallel alignment. Color scale denotes frequency of calculated trajectory.

~10 nm and ~5 nm, respectively, and the barrier is ~1 kT. The cDFT calculations show that the combination of electrostatic, solvation, ion correlation and van der Waals terms establish an interaction potential that is attractive at short range (~2 nm) with little or no barrier to attachment (≤ ~1 kT), while the Langevin dynamics simulations show that the dipole–dipole forces and torques arising from the inherent dipole of ZnO provide attractive forces and aligning torques at long range (> ~10 nm) and give an interaction potential vs particle separation in good agreement with that extracted from the experimental RDFs. Thus, this combination of short and long range interactions is adequate to explain the behavior of ZnO nanoparticles. However, there are other effects that may play a role, even if only secondarily.

The first effect is the electron beam, which was shown to promote the aggregation of Au colloids[28]. However, in that case, the colloidal dispersion itself was stable and the creation of solution radicals by the beam, which then mitigated the repulsive electrostatic interactions, destabilized the dispersion. In our case, the particles assemble by OA whether or not an electron beam is present. Moreover, electron beam effects on the particles themselves are unlikely to create either the long-range attractive interaction or torques, because any induced charges or dipoles would be of like sign and thus be repulsive.

The second potential effect stems from the prediction of the full cDFT model of long-range orientation-dependent oscillations in the interparticle potential. That prediction suggests a picture in which the specific crystallographic face is only important when a non-electrostatic solvation term is present in the model. This sensitivity to the details of solvation is interesting and could provide a competitive driving force for assembly. While we leave this open as a possibility, significant research is required to further test this hypothesis. Given that solvation is sensitive to ion-solvent correlations that are currently not present in the description provided by the non-electrostatic mean field approach, our attempt at capturing these effects, even at the mean field level, points to potentially interesting driving forces that cannot be explained with primitive models (see Fig. 5d and Supplementary Fig. 9). Because these long-range oscillations are directly related to solution structuring and are difficult to test and validate with either molecular simulations or experiments that probe forces between crystal faces indirectly, direct measurements, such as those made with AFM-based force spectroscopy using single-crystal tips[7,12,32,64], can provide an important test.

A third potential modifying influence may arise from anomalous dispersion forces arising from the close proximity of the particles to the membrane of the TEM fluid cell. Similar long-range particle interactions were observed in a recent in situ TEM study of Ag nanoparticles[65], suggesting the common feature of confinement may drive these particle interactions. Confinement near either dielectric or conducting surfaces has been predicted to alter the fluctuations of the electric field in the proximity of two particles, in a complex non-additive manner conceptually similar to that associated with interactions between three bodies. The presence of a dielectric interface was predicted to create a more slowly-decaying interaction potential between two atoms, scaling as $1/z^3$, that adds to the bulk term, which scales as $1/z^6$, where $z$ is a distance between atoms[66]. Furthermore, confinement is predicted to greatly enhance the dispersion interaction between point-like particles by ~ two orders of magnitude, depending on the geometry and dielectric characteristics of the interface[67]. These studies suggest that confinement alone can lead to long-range attractive dispersion interactions. Electrostatics would add additional complexity to the slowly-decaying nature of the interaction. Because the inherent anisotropic dielectric response of crystals is known to produce a torque on two approaching crystals[7], this confinement-induced enhancement could also lead to an anomalous long-range torque. The magnitude of this effect is dependent on the dielectric response of the confining substrate, confinement geometry, and polarizability of the particles; a further study with various systems and salt concentrations is needed to test this postulation. For example, a *cryo*-TEM study of the distributions and orientations of many particles captured from bulk solution could provide that test. If the confinement effect was the source of the long-range interaction, then the measured $W(h)$ should be featureless down to smaller separations than seen here and the particles should be largely misaligned above ~ 2 nm separation and attached on a variety of faces.

The results presented here provide a number of new insights into the dynamics of OA, as well as the underlying forces that are responsible for both aggregation and alignment. First, these results show that, even in regimes of extremely slow diffusion, attractive forces and torques due to interparticle potentials are sufficiently large to overcome drag forces, as well as Brownian forces and torques. Moreover, these attractive forces and torques can act at particle separations much larger than expected from a simple Derjaguin–Landau–Verwey–Overbeek (DLVO) picture or from standard molecular simulations and can lead particles to reach coalignment well before they reach a distance where strong attractive potentials drive the final jump to contact. This conclusion contrasts with that derived from previous studies of dipole-free systems, where alignment is attributed to short range interactions[7,9,27].

While a number of studies have used in situ TEM to investigate oriented attachment of nanoparticles in aqueous solutions, the extraction of potentials and simulation of interaction forces has rarely been applied and then only to metallic systems, most notably

gold functionalized with organic ligands. A recent study, which extracted the interparticle potential for citrate-functionalized gold nanoparticles from radial distribution functions, found that the range of the attractive interaction was only about 2 nm and the potential exhibited both a distinct solvent separated minimum and a significant barrier to attachment due to the surface-bound ligands[28]. These results stand in stark contrast to those of the current study on ZnO, which, unlike the gold system, possesses an electric dipole and has no ligands. Because oxides are a ubiquitous class of materials in both natural and synthetic environments, and because dipole moments are a characteristic of many non-metallic particles, the results presented here for the ZnO system provide both an important counterpoint to previous studies on ligand-functionalized metals, as well as conclusions that should be broadly applicable to a vast array of crystal systems in natural and laboratory settings.

## Methods

**Synthesis**. Zinc acetate dihydrate (0.01 M) was dissolved in methanol under vigorous stirring at about 60 °C. Subsequently, a 0.03 M solution of KOH in methanol was added dropwise at 60 °C. The reaction mixture was stirred for 2h at 60 °C. The resulting solution was cleaned three times with methanol and water and, finally, was dispersed in methanol.

**TEM experiments**. The static liquid cell (Hummingbird Scientific, USA) design was based on microfabricated silicon chips. Two square silicon chips containing silicon nitride membranes were separated with 100 nm gold spacers deposited on the two strips. Prior to loading the solution, the membranes were oxygen plasma cleaned for 1 min to remove organic contaminants and render the surfaces hydrophilic. The solution was deposited between the two membranes in 0.8 μL droplets placed onto the bottom side of the membrane followed by capping with an upper chip. The liquid cell was mounted and sealed on the holder using O-rings. The assembled liquid-cell stage was vacuum tested in a Pfeiffer vacuum chamber. All in situ observations and ex situ characterization were performed on a Tecnai F20 (200 keV, FEI, USA). The images were recorded using an FEI Eagle charge-coupled device (CCD) camera. Continuous movies were captured using a freeware screen grabber (VirtualDub) with a time resolution of about 2 s, which is limited by the CCD camera.

**Image analysis**. The ZnO nanoparticles were detected and tracked using in-house MATLAB algorithms. Specifically, the images were passed through a median filter and binarized using a suitable threshold value. The center of mass and area of each particle were then calculated based on pixels forming continuous, distinct objects. Since the particles move relatively slowly, each particle was easily matched to the closest particle in the previous frame. Particle orientation was determined using frame-by-frame Fourier transform analyses.

**Theoretical methods**. To understand the thermodynamic driving forces, a cDFT-based atomistic-to-mesoscale approach was developed[52,68]. The details of cDFT formulation used in this work were described in detail elsewhere[52] and also provided in the Supplementary Information for completeness. The methanol electrolyte solution was modeled as a dielectric medium with $\varepsilon = 32.7$, consisting of discrete charged spherical particles representing ions, and neutral spherical particles representing methanol molecules. The concentration of spherical "methanol molecules" was 35 M, chosen to model experimental methanol density. We used experimental crystalline ionic diameters for mobile ions equal to 0.15 nm for $Zn^{2+}$ ions and 0. 522 nm for $Ac^-$ ions, and a van der Waals diameter equal to 0.285 nm for methanol molecules[69,70]. The ion charges were equal to $+2$ for $Zn^{2+}$ ions, $-1$ for $Ac^-$ ions, and 0 for methanol. Short-range interactions were treated explicitly. Experimental enthalpies of solvation were used for short-range interactions of ions with methanol. cDFT simulations account only for average chemical interactions between ion and solvent species with the surface and between the species, e.g. solvation interactions, which limits the applicability of the approach to edge-to-edge interparticle separations of no smaller than 0.5 nm. However, as reported in our previous publications, the accuracy of our cDFT model in calculating thermodynamic properties of electrolyte solutions, e.g. ion activity in a wide concentration range, and ion distribution at complex interfaces exceeds that of all-atom molecular dynamics with standard force-fields and the calculated results are in close agreement with state-of-the-art experimental data[71,72] rendering the methodology adequate for the task.

## Data availability

The datasets generated during and/or analyzed during the current study are available from the corresponding author on reasonable request.

## Code availability

The simulation code generated during and/or analyzed during the current study are available from the corresponding author on reasonable request.

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

## Acknowledgements

This work was supported by the US Department of Energy (DOE), Office of Basic Energy Sciences (BES) at Pacific Northwest National Laboratory (PNNL). All experimental work, analysis of particle aggregation kinetics and hydrodynamic forces, and evaluation of confinement effects were supported by the BES Division of Materials Science and Engineering, Synthesis and Processing Sciences Program. cDFT simulations and analysis of dipolar interactions were supported by the BES Chemical Sciences, Geosciences, and Biosciences Division, Geosciences Program and Chemical Physics and Interfacial Sciences Program, respectively. Simulations were performed using PNNL Institutional Computing resources. A preliminary concept on the confinement effect was partially supported by the Laboratory Directed Research and Development Program at PNNL. PNNL is a multiprogram national laboratory operated for DOE by Battelle under Contract No. DE-AC05–76RL01830. L. Liu wishes to express her many thanks to Dr. Dongsheng Li for training in in-situ TEM techniques. The authors also acknowledge Dr. Xin Zhang and Dr. Zhizhang Shen in the Geochemistry group for helpful discussions about ZnO crystal growth mechanism and theories, Xinran Liang for drawing the scheme, and Dr. Benjamin A. Legg for useful advice on image analysis.

## Author contributions

All authors contributed to the writing of the paper. L.L. performed the experiments. E.N. and L.L., along with J.J.D.Y. and J.C., performed the data analyses and interpretation. J.C. and C.J.M. developed the concepts and formulations for 2D-based particle diffusivity and aggregation kinetics. M.L.S. developed the classical density functional theory model. G.K.S. and J.J.D.Y. developed the concepts for dipole–dipole interactions and G.K.S. performed the calculations. J.C. and J.J.D.Y. designed the study.

## Competing interests

The authors declare no competing interests.
