## [Peer Review File · Nature Communications]

Reviewers' comments:

Reviewer #1 (Remarks to the Author):

Attachment and rotation of adjacent particles to find a crystallographic alignment is a key step of OA growth. Subsequently, a jump-to-contact process results in final growth. Discussing the driving force of OA growth mechanism is a key to tailoring the OA growth. In literatures including authors' previous paper published in ACS Nano 2014, 8,7,6526, it is a common sense that the dipole-dipole interaction and solvent-surface interactions could play critical roles. The distance of action and driving force have also been discussed by many other similar work based on real-time TEM observations. The most interesting findings of presenting work is a distinct length scale of attractive interactions and torques seen in the experiments (~10 nm and 5 nm), from those predicted by theory (< 3 nm). However, it remained unclear if a high concentration of ZnO particles in design experiment restricts the diffusion of particles in limited range. The diffusion of particles will be confined in limited environment, owing to a steric hindrance and repulsive force coming from adjacent particles. As a result, it looks like there is an attraction of adjacent particles in large length. In my opinion, these factors had better be taken into account when discussing the length scale of particles attachment. Furthermore, the authors showed that OA is driven by forces and torques due to a combination of electrostatic ion-solvent correlations and dipolar interactions that act at separations well beyond 5 nm. It can be speculated that the choice of particle size could significantly impact the predicting result. When the particle size is reduced to several nanometers, the number of surface atoms will be significantly distinct from particles with different crystalline sizes. The surface structure always determines the surface on which attachment occurs, and the value of the force as well. It seems most of evidence was about the distance investigation without clear structural analysis. Finally, it seem the presenting paper still have big weakness in proposing new and strong evidence on the driving force of OA growth, and on how to modulate the OA growth. As aforementioned, it has been proposed the dipole-dipole interaction and solvent-surface interactions contribute to the OA growth. There is no solid evidence here for answering what impacts the attachment and rotation of particles, and what enables the particles to achieve crystallographic alignment. All these questions are key for achieving a fine tailoring over the OA growth.

Reviewer #2 (Remarks to the Author):

Based on some liquid-phase in situ TEM observation on how the ZnO nanoparticles attached to each other, authors gives some theoretical explanation of the mechanism of the oriented attachment (OA) of

ZnO nanoparticle. I have no background on the theoretical calculation, so most of my comments will be focused on experimental part.

1) Beside the crystal attachment at the $\pm(0001)$ plane, any attachment from other planes had been identified? the planar distance of (0002) plane is $\sim 2.8\text{\AA}$, while $(01-10)$ is $\sim 2.6\text{\AA}$ and $(01-11) \sim 2.5\text{\AA}$. they all should above the resolution limits of the microscope.

2) More explanation is needed when authors discussed the Fig. 2g. How the 1D intensity profiles generated. What the color stands for? If the yellow color means the C direction, why there has a distribution range at some certain time? where is the zero degree orientation?

3) please specify the A1 and A2 in page 7.

4) in $g(h)$ definition, what the Δh stands for? in the paper authors cited, the r (here is h) is the absolute distance, whilst here h is edge-to-edge particle separation (h). Can they still use the h to replace the r in the original definition of $g(r)$?

5) In the figure 4 caption, the first "(b)" should be "(a)". Also in Fig. 4, What the difference among the different colored curves in (c)? which curve stands for the 2.2 nm particle?

6) In S6, the rang of h only covered 0-6 nm, while in Fig. 4(a) and (b), it goes from 0-30 nm. can authors supply an extra overview range plot same as the S6 one?

7) The discussion, it confused me that authors discussed three possibilities, but they seems didn't rule out any of them to leave it as an open questions.

Reviewer #3 (Remarks to the Author):

Review of "Connecting energetics to dynamics in particle growth by oriented attachment using real-time observations" by Liu et al.

In this paper, the authors describe experimental work on the aggregation of ZnO into quasi-one dimensional structures in solution. Time resolved TEM imaging shows the dominant mechanism to be one of aggregation of nanoparticles into chains rather than monomer-attachment. The authors carefully characterized the nanoparticles and in a few (3) cases, were able to confirm that they seemed to align prior to attachment so as to form a single crystal, rather than grain boundaries. The alignment necessarily involves rotation, and so a torque, which seems to involve interactions when the clusters are up to 10nm apart. Significant effort then went into trying to understand the origin of such long-ranged forces with several possibilities considered (electrostatic forces, the quasi 2D TEM cell or fluid structural effects) but with no definitive conclusion.

This paper describes very careful and beautiful experimental work and certainly merits publication in Nature Communications. However, there are several points that should be clarified in order to aid in understanding the work.

1. I found the images in Fig. 2g difficult to interpret - perhaps more could be said about how they reveal the orientation of the nanoparticles.

2. The cDFT calculations play an important role in interpreting the experimental results but the description of the calculations is hurried with many details missing. Specifically, I would like to know the following:

a) On page 14 of the SI, in the Poisson equation, what is ρf ?

b) What is the explicit expression, particularly the density dependence, for F_{eXDL} , which is to say, what are the

expressions for F_c^{eX} and F_{eX}

im?

c) What is the explicit expression, particularly the density dependence, for F_{eX} ion VDW ?

d) What was the size of the computational cell and its boundary conditions? Did the size vary from one calculation to another?

e) What numerical method was used. If it is finite elements, what was the lattice spacing? If pseudo-spectral, what were the basis functions? I realize that it would not be appropriate to give all details but a general description of the method would be helpful.

f) Were the calculations performed at constant particle number or chemical potential? If at constant particle number, how was the number chosen in each calculation? If at constant chemical potentials, how were those chosen?

3. I note that on the last page of the SI, the text reads : "Density profiles are calculated within cDFT via the minimization of the excess free energy..." - this must be a typo as one minimizes the total (not just the excess) free energy functional.

4. How were the disjoining pressures calculated? I suppose it was by varying the separation of the particles and then differentiating the free energy with respect to the distance between faces. Is that correct?

5. The large oscillations in the pressure at large separations are quite puzzling, as noted by the authors. It would be helpful if they would show the corresponding density profiles. Do they also show such large oscillations? Can the oscillations be suppressed by turning off one of the terms, e.g. the Coulombic term in the energy?

6. How was the interaction potential in figure S9 calculated?

Reviewer #1 (Remarks to the Author):

Comment: Attachment and rotation of adjacent particles to find a crystallographic alignment is a key step of OA growth. Subsequently, a jump-to-contact process results in final growth. Discussing the driving force of OA growth mechanism is a key to tailoring the OA growth. In literatures including authors' previous paper published in ACS Nano 2014, 8,7,6526, it is a common sense that the dipole-dipole interaction and solvent-surface interactions could play critical roles. The distance of action and driving force have also been discussed by many other similar work based on real-time TEM observations.

Response: We agree with the reviewer that solvent-surface interactions and dipole-dipole interactions are obvious forces to consider, but whether or not they turn out to be the forces that dominate or whether electrostatic and/or van der Waals interactions dominate has not been resolved in past literature. In fact, although, as the reviewer points out, there are numerous studies that have used *in situ* TEM to examine particle aggregation events, only one other study has collected statistics on particle distributions to extract the interaction potential and combined that with state of the art simulations to determine the underlying force components. That excellent study, which was published in this journal and cited in our manuscript (Zhu et al., *Nature Comm.* **2018**, 9, 421), was performed on gold for which dipole-dipole interactions are not a factor and revealed a solvent separated minimum with a significant barrier opposing the jump to contact, neither of which are seen in our data for ZnO. Moreover, the gold particles were functionalized with surface ligands, which limits the extent to which the conclusions can be extended to bare inorganic particles. These differences render our results all the more significant, because they show that individual particle systems can exhibit dramatic differences. Because oxides are a ubiquitous class of materials in both natural and synthetic environments, and because dipole moments are a common characteristic of non-metallic particles, our work provides both an important counterpoint to that previous study and conclusions that are broadly applicable.

To address this comment in the manuscript, we have added the following to the end of the Discussion:

“While a number of studies have used *in situ* TEM to investigate oriented attachment of nanoparticles in aqueous solutions, the extraction of potentials and simulation of interaction forces has rarely been applied and then only to metallic systems, most notably gold functionalized with organic ligands. A recent study, which extracted the interparticle potential for citrate-functionalized gold nanoparticles from radial distribution functions, found that the range of the attractive interaction was only about 2 nm and the potential exhibited both a distinct solvent separated minimum and a significant barrier to attachment due to the surface-bound ligands. These results stand in stark contrast to those of the current study on ZnO, which, unlike the gold system, possesses an electric dipole and has no ligands. Because oxides are a ubiquitous class of materials in both natural and synthetic environments, and because dipole moments are a characteristic of many non-metallic particles, the results presented here for the ZnO system provide both an important counterpoint to previous studies on ligand-functionalized metals, as well as

conclusions that should be broadly applicable to a vast array of crystal systems in natural and laboratory settings.”

Comment: The most interesting findings of presenting work is a distinct length scale of attractive interactions and torques seen in the experiments (~10 nm and 5 nm), from those predicted by theory (< 3 nm). However, it remained unclear if a high concentration of ZnO particles in design experiment restricts the diffusion of particles in limited range. The diffusion of particles will be confined in limited environment, owing to a steric hindrance and repulsive force coming from adjacent particles. As a result, it looks like there is an attraction of adjacent particles in large length. In my opinion, these factors had better be taken into account when discussing the length scale of particles attachment. Furthermore, the authors showed that OA is driven by forces and torques due to a combination of electrostatic ion-solvent correlations and dipolar interactions that act at separations well beyond 5 nm. It can be speculated that the choice of particle size could significantly impact the predicting result. When the particle size is reduced to several nanometers, the number of surface atoms will be significantly distinct from particles with different crystalline sizes. The surface structure always determines the surface on which attachment occurs, and the value of the force as well. It seems most of evidence was about the distance investigation without clear structural analysis.

Response: We thank the reviewer for recognizing that a key finding of this study is the length scale of the attractive interactions. The reviewer raises two concerns about the validity of our conclusions. The first is that restricted motion is responsible for making it look like forces act at a distance. We believe we have explicitly addressed this in our manuscript, but perhaps we did not make a clear enough statement about it. In our manuscript, we provided an extensive theoretical analysis of the impact of the restricted motion that arises from the high shear viscosity that exists within nm-scale distances from surfaces — in this case the liquid cell membrane — and this analysis is the first ever presented despite there being a decade worth of liquid cell TEM publications. For example, the study by Zhu et al. referred to above does not provide this analysis. In fact, we considered publishing that aspect of the work separately, because the entire liquid cell TEM community is analyzing data absent an understanding of this effect. Through that analysis we show that the impact of this restriction is to render the aggregation kinetics mass transport limited. *But this is not a limitation of the work, it is a positive feature*, because it enables us to extract an *equilibrium* radial distribution function (RDF) and thus determine the magnitude of the barrier and make direct comparisons to simulations. More importantly for the purposes of allaying the reviewer’s concern, these analyses show that the range of the potential and existence of an aligning torque are unrelated to the extent to which particle motion is restricted or unrestricted. There is simply no physical connection between the viscous limitation on transport and the potential that drives that transport. The former just determines the rate of response to the latter. Furthermore, our experimental conditions do not lead to extremely high particle concentrations that could potentially be responsible for the steric hindrance and repulsive force to which the reviewer refers and the fact that we are not in that regime is demonstrated by the radial distribution function extracted from positions of the particles. If such repulsive interactions existed, they would be revealed by the RDF. Moreover, the RDF

would then be evolving in time as particles merged and that is not observed. Thus, there is no reason why the effect of confinement either by the cell or the presence of other particles should call into question the key conclusions of the study.

The second point of concern raised by the reviewer is that the forces are affected by particle size. They are and, indeed, they are shown to be so in our simulations and theoretical analyses, but we fail to see how that is relevant to the validity or value of the study. After all, we used particles that have dimensions completely typical for nanoparticle synthesis. Moreover, they are typical for domain sizes seen in nanocrystals grown by oriented attachment and the reason for that is fairly intuitive: sub-nm particles suffer little penalty by being misaligned and particles that are hundreds of nm in size cannot overcome the hydration barriers required for oriented attachment. Moreover, because our calculations and simulations include the effect of particle size, they are easily applied to any dimension of interest. We fail to see why choosing a typical and relevant size to do experiments that are then analyzed with a formalism that accounts for size is even a limitation of the work, let alone a basis for rejection. That would then be a basis for rejecting every liquid cell TEM paper that has ever been published on nanoparticle dynamics.

To address this comment in the manuscript, we have added the following on page 8:

“A frame-by-frame analysis shows that the RDF did not change appreciably during the course of the experiment, and hence provides representative information about the quasi-steady state distribution of particle separations (Figure S6). Moreover, the constancy of the RDF implies that particle concentrations are sufficiently low that steric hindrance and repulsive forces due to neighboring particles are negligible.”

On page 10 we added:

“Since the stability ratio can span as many as six orders of magnitude for a single colloidal system, depending on the surface and solution chemistry,⁴³ this low-to-intermediate value further supports the conclusion of a negligible energy barrier for particle attachment and thus a diffusion-limited process. This analysis also shows that the effect of the membrane is simply to impose a high viscosity that resists particle motion, thus rendering it mass transport limited. As a consequence, the extracted radial distribution function represents a quasi-equilibrium distribution.”

Comment: Finally, it seem the presenting paper still have big weakness in proposing new and strong evidence on the driving force of OA growth, and on how to modulate the OA growth. As aforementioned, it has been proposed the dipole-dipole interaction and solvent-surface interactions contribute to the OA growth. There is no solid evidence here for answering what impacts the attachment and rotation of particles, and what enables the particles to achieve crystallographic alignment. All these questions are key for achieving a fine tailoring over the OA growth.

We are surprised by this comment, because we feel that a major strength of our paper is the solid nature of the data, analyses and simulations. The positions of the particles and the extracted orientations are clear, the analysis of the data is straight forward and the conclusions are unambiguous: the particles are attracted and experience an aligning torque starting from at separations of ~10 nm and ~5 nm, respectively, and the barrier is ~1 kT. The cDFT calculations include the electrostatic, solvation, ion correlation and van der Waals terms for the precise material system of the experiments, and the dipole-dipole forces and torques are calculated for same system and used to predict particle motion in the medium of similar viscosity using a well-accepted Langevin dynamics approach. The results of the calculations and simulations are clear: the cDFT results show that ion correlation forces lead to the short range attractive interaction and elimination of a solvation barrier and the Langevin dynamics simulations show that the dipole-dipole interactions reproduce the potential vs separation extracted from the radial distribution function. In short, this study goes well beyond any that have come before it — with the exception of the paper by Zhu et al. referred to above — in giving conclusive evidence for the forces that drive OA.

Perhaps the manner in which we presented the analysis of dipole-dipole interactions created a source of confusion about the certainty of the conclusions, in that we listed it in a set of three possibilities. To avoid that, we have now moved that analysis to come right after the section on cDFT and added a new introductory paragraph to the discussion section.

The transition from the section on cDFT to the section on dipole forces now reads:

“Thus, we conclude that ZnO OA along [0001] dominates because long range torques pre-align the particles before they reach separations at which jump to contact can occur.

The source of the long range attractive interaction and aligning torque can be understood when the inherent electric dipole of ZnO is taken into account. While the cDFT model presented above incorporates the electrostatic, van der Waals, solvation and ion correlation forces, it does not include dipole-dipole forces, which must exist, given that ZnO is a polar material presenting Zn and O terminations, respectively, at the [0001] and [000-1] faces, leading to a spontaneous dipole moment.⁶⁴”

The Discussion section now starts with:

“The findings reported above show that ZnO nanoparticles are attracted and experience an aligning torque starting from at separations of ~10 nm and ~5 nm, respectively, and the barrier is ~1 kT. The cDFT calculations show that the combination of electrostatic, solvation, ion correlation and van der Waals terms establish an interaction potential that is attractive at short range (~2 nm) with little or no barrier to attachment (\leq ~1kT), while the Langevin dynamics simulations show that the dipole-dipole forces and torques arising from the inherent dipole of ZnO provide attractive forces and aligning torques at long range (>~10nm) and give an interaction potential vs particle separation in good agreement with that extracted from the experimental RDFs. Thus this combination of short and long range interactions is adequate to explain the behavior of ZnO

nanoparticles. However, there are other effects that may play a role, even if only secondarily.”

Each of the possibilities in the Discussion section now start with appropriate sentences: “The effect first is the electron beam,...”, “The second potential effect stems from the prediction of the full cDFT model...”, and “A third potential modifying influence may arise from anomalous dispersion forces arising from the close proximity of the particles to the membrane...”

Reviewer #2 (Remarks to the Author):

Based on some liquid-phase in situ TEM observation on how the ZnO nanoparticles attached to each other, authors gives some theoretical explanation of the mechanism of the oriented attachment (OA) of ZnO nanoparticle. I have no background on the theoretical calculation, so most of my comments will be focused on experimental part.

1) Beside the crystal attachment at the $\pm(0001)$ plane, any attachment from other planes had been identified? the planar distance of (0002) plane is $\sim 2.8\text{\AA}$, while $(01-10)$ is $\sim 2.6\text{\AA}$ and $(01-11) \sim 2.5\text{\AA}$. they all should above the resolution limits of the microscope.

In the in-situ experiment, we only observed particle attachment along the $\pm(0001)$ plane, but also observed in ex-situ experiments rare cases of attachment along (100) , as shown in Figure 1b. It is worth noting, as we do in the manuscript, that the cDFT calculations do not preclude other attachment configurations. The rationale for why we only see $\pm(0001)$ attachment events is that the dipole-dipole interactions act at sufficiently far distance to align the particles along the $\pm(0001)$ planes prior to their collision and attachment.

Indeed, the lattice spacings of $\sim 2.8\text{\AA}$, 2.6\AA and 2.5\AA are very close to each other, but still above the resolution limits of the microscope. Thus, the in situ data clearly show that ZnO rod attachment along $[0001]$ is preferable and this is unambiguously supported by the ex situ data, which clearly show that growth along $[0001]$ overwhelmingly dominates, as shown in the Figure S1(D) and (E) below. To address this comment in the manuscript, we have added more images in Figure S1 in the SI.

2) More explanation is needed when authors discussed the Fig. 2g. How the 1D intensity profiles generated. What the color stands for? If the yellow color means the C direction, why there has a distribution range at some certain time? where is the zero degree orientation?

The panels in Figure 2g plot the evolution of the orientation of particles I, II, and the final particle after attachment, respectively. At each time point, a 1D profile is obtained from a Fourier transform analysis of the relevant particle (comparable to Figure S4), by summing the intensity from the FT data at each angle θ . In Figure 2g, yellow color denotes high intensity.

In fact, there is no distribution range in the data. The reason why the data is noisy is because the lattice fringes are not obvious in every frame of the TEM movie. Despite this, our analysis can track the relative particle orientation as the trace of yellow intensity in the three panels.

For more clarity, we have adjusted the data such that $\theta = 0$ denotes particle alignment, instead of an arbitrary frame of reference.

We have clarified the explanation of Figure 2g in the caption:

“1D intensity profiles versus time created from FFT analyses of image sequence show that the two particles rotated and aligned their crystallographic axes within 20 seconds prior to contact. This result is manifested as a trace of high contrast (yellow) in the panels of Figure 2g.”

3) please specify the A1 and A2 in page 7.

The discussion on A1 and A2 refers to all particle pairs that eventually attach (not one specific particle). The corresponding Figure 3e plots the area evolution of six particle pairs. The main message is that the analysis of the individual areas before attachment and the final area after attachment shows that the total particle volume is constant, i.e. we do not observe any particle growth except by attachment.

To clarify this point in the manuscript, we have modified the text on page 5 to read,

“To evaluate the contribution to particle growth from the attachment vs ion addition, we derived the expected dependence of projected particle area on volume for the case where the volume of the final particle is simply given by the sum of the volume of the two primary particles. The evolution of the resulting normalized particle area defined as $A_{\text{nor}} = 1/(A_1^{3/2} + A_2^{3/2})^{2/3}$ (Figure 3a-c; derived formula shown in SI) between attaching particles (Figure 3e) shows that the volume of the newly merged particle consisted approximately of the sum of the volumes of the two individual particles.”

We have also modified the figure to highlight the average particle area before and after attachment and added the following statement to the caption of Figure 3:

“The grey dotted lines denote the average particle size prior to and after attachment.”

4) *in g(h) definition, what the delta-h stands for? in the paper authors cited, the r (here is h) is the absolute distance, whilst here h is edge-to-edge particle separation (h). Can they still use the h to replace the r in the original definition of g(r)?*

Our analysis is indeed in terms of the edge-to-edge particle separation “h”, which does not change the physical interpretation of g(h). Note that “h” is a more useful parameter in our system since the particles have a non-uniform size, and hence calculations based on “r” would lead to mis-interpretation of the data.

In this context, $\Delta h = 0.2$ nm. We have added this information to the manuscript text on page 8:

“To determine the underlying free energy landscape across which particles diffuse and interact, we analyzed the collective behavior of particle ensembles (Figure 3a-c; Movie 4) by calculating the radial distribution probability function (RDF) (Figure 4a) according to $g(h) = \frac{1}{2\pi h \Delta h} \frac{N_h}{\rho}$, where N_h is the number of particles within $[h-\Delta h/2, h+\Delta h/2]$, $\Delta h = 0.2$ nm, and ρ is the number of particles per area unit.”

5) *In the figure 4 caption, the first "(b)" should be "(a)". Also in Fig. 4, What the difference among the different colored curves in (c)? which curve stands for the 2.2 nm particle?*

We thank the reviewer for pointing out the typo in the figure caption, which is now corrected.

2.2 nm represents the average particle size. For example, the blue and magenta data points represent particles of radius 2.1 nm. Note that the key result from this analysis is that the order of magnitude of the diffusion coefficient is six times lower than expected from the Stokes-Einstein relationship, which is evident from the MSD data.

We modified the relevant sentence on page 9 of the manuscript accordingly:

“We then calculated mean squared displacements for multiple particles (Figure 4c) and obtained an average diffusion coefficient of $2.99 \text{ nm}^2/\text{s}$ for particles of radius $a = 2.2$ nm, which is the average particle radius in the experiments (see SI for details);...”

6) *In S6, the range of h only covered 0-6 nm, while in Fig. 4(a) and (b), it goes from 0-30 nm. can authors supply an extra overview range plot same as the S6 one?*

Based on the reviewer's comment, we have expanded the range of Figure S6 to 30 nm.

7) The discussion, it confused me that authors discussed three possibilities, but they seems didn't rule out any of them to leave it as an open questions.

We apologize for the confusion. We made a poor choice in listing dipole-dipole interactions as one of the three possibilities. Our simulations show that dipole-dipole interactions alone account for the length scale and strength of the long-range interactions, regardless of whether either of the other two phenomena are present as second order effects. Even though we stated this in the section on dipole-dipole interactions, the placement of that section following the statement of three possibilities made it seem like there is an open question as to whether the dipole-dipole interaction accounts for long range forces and torques. Consequently, we have revised the manuscript by moving that section so that it follows the cDFT analysis. The discussion of the other two possibilities is now presented as other effects that may modify the potential determined from cDFT and dipole-dipole interactions. To address this problem, we have now moved that analysis to come right after the section on cDFT and added a new introductory paragraph and final paragraph to the discussion section.

The transition from the section on cDFT to the section on dipole-dipole interaction now reads:

“Thus, we conclude that ZnO OA along [0001] dominates because long range torques pre-align the particles before they reach separations at which jump to contact can occur.

The source of the long range attractive interaction and aligning torque can be understood when the inherent electric dipole of ZnO is taken into account. While the cDFT model presented above incorporates the electrostatic, van der Waals, solvation and ion correlation forces, it does not include dipole-dipole forces, which must exist, given that ZnO is a polar material presenting Zn and O terminations, respectively, at the [0001] and [000-1] faces, leading to a spontaneous dipole moment.⁶⁴”

The Discussion section now starts with:

“The findings reported above show that ZnO nanoparticles are attracted and experience an aligning torque starting from at separations of ~10 nm and ~5 nm, respectively, and the barrier is ~1 kT. The cDFT calculations show that the combination of electrostatic, solvation, ion correlation and van der Waals terms establish an interaction potential that is attractive at short range (~2 nm) with little or no barrier to attachment ($\leq \sim 1\text{kT}$), while the Langevin dynamics simulations show that the dipole-dipole forces and torques arising from the inherent dipole of ZnO provide attractive forces and aligning torques at long range ($> \sim 10\text{nm}$) and give an interaction potential vs particle separation in good agreement with that extracted from the experimental RDFs. Thus this combination of short and long range interactions is adequate to explain the behavior of ZnO nanoparticles. However, there are other effects that may play a role, even if only secondarily.”

Each of the possibilities in the Discussion section now start with appropriate sentences: “The effect first is the electron beam,...”, “The second potential effect stems from the prediction of the full cDFT model...”, and “A third potential modifying influence may arise from anomalous dispersion forces arising from the close proximity of the particles to the membrane...”

The Discussion section now ends with:

“...Moreover, these attractive forces and torques can act at particle separations much larger than expected from a simple DLVO picture or from standard molecular simulations and can lead particles to reach coalignment well before they reach a distance where strong attractive potentials drive the final jump to contact. This conclusion contrasts with that derived from previous studies of dipole-free systems, where alignment is attributed to short range interactions.

While a number of studies have used *in situ* TEM to investigate oriented attachment of nanoparticles in aqueous solutions, the extraction of potentials and simulation of interaction forces has rarely been applied and then only to metallic systems, most notably gold functionalized with organic ligands. A recent study, which extracted the interparticle potential for citrate-functionalized gold nanoparticles from radial distribution functions, found that the range of the attractive interaction was only about 2 nm and the potential exhibited both a distinct solvent separated minimum and a significant barrier to attachment due to the surface-bound ligands. These results stand in stark contrast to those of the current study on ZnO, which, unlike the gold system, possesses an electric dipole and has no ligands. Because oxides are a ubiquitous class of materials in both natural and synthetic environments, and because dipole moments are a characteristic of many non-metallic particles, the results presented here for the ZnO system provide both an important counterpoint to previous studies on ligand-functionalized metals, as well as conclusions that should be broadly applicable to a vast array of crystal systems in natural and laboratory settings.”

Reviewer #3 (Remarks to the Author):

In this paper, the authors describe experimental work on the aggregation of ZnO into quasi-one dimensional structures in solution. Time resolved TEM imaging shows the dominant mechanism to be one of aggregation of nanoparticles into chains rather than monomer-attachment. The authors carefully characterized the nanoparticles and in a few (3) cases, were able to confirm that they seemed to align prior to attachment so as to form a single crystal, rather than grain boundaries. The alignment necessarily involves rotation, and so a torque, which seems to involve interactions when the clusters are up to 10 nm apart. Significant effort then went into trying to understand the origin of such long-ranged forces with several possibilities considered (electrostatic forces, the quasi 2D TEM cell or fluid structural effects) but with no definitive conclusion.

This paper describes very careful and beautiful experimental work and certainly merits publication in Nature Communications. However, there are several points that should be clarified in order to aid in understanding the work.

1. I found the images in Fig. 2g difficult to interpret - perhaps more could be said about how they reveal the orientation of the nanoparticles.

The panels in Figure 2g plot the evolution of the orientation of particles I, II, and the final particle after attachment, respectively. At each time point, a 1D profile is obtained from a Fourier transform analysis of the relevant particle (comparable to Figure S4), by summing the intensity from the FT data at each angle θ . Yellow contrast denotes high intensity. Importantly, the spread in yellow pixels does not indicate there is a distribution range in the values of θ . The reason why the data is noisy is because the lattice fringes are not obvious in every frame of the TEM movie. Despite this, our analysis can track the relative particle orientation as the trace of yellow intensity in the three panels. For more clarity, we have adjusted the data such that $\theta = 0$ denotes particle alignment, instead of an arbitrary frame of reference, and have modified the caption to Figure 2g to read:

“1D intensity profiles versus time created from FFT analyses of image sequence show that the two particles rotated and aligned their crystallographic axes within 20 seconds prior to contact. This result is manifested as a trace of high contrast (yellow) in the panels of Figure 2g.”

2. The cDFT calculations play an important role in interpreting the experimental results but the description of the calculations is hurried with many details missing. Specifically, I would like to know the following:

a) On page 15 of the SI, in the Poisson equation, what is ρ_f ?

Response: We clarified the notations used in the Poisson equation by adding: “..where $\rho_f(\mathbf{r})$ is the fixed charge density on nanoparticle facets, $\rho_i(\mathbf{r})$ is the density of mobile species with charge q_i ,...”

b) What is the explicit expression, particularly the density dependence, for $F^{\text{ex_EDL}}$, which is to say, what are the expressions for $F^{\text{ex_C}}$ and $F^{\text{ex_im}}$?

Response: The direct Coulomb term can be calculated exactly giving the following free energy contribution:

$$F_C^{\text{ex}} = \frac{kTl_B}{2} \sum_{i,j=p,+,-} \iint \frac{q_i q_j \rho_i(\mathbf{r}) \rho_j(\mathbf{r}')}{|\mathbf{r} - \mathbf{r}'|} d\mathbf{r} d\mathbf{r}';$$

where the Bjerrum length is defined as $l_B = \frac{e^2}{kT\epsilon}$. However, only expressions are available for the image contribution. For this reason, Coulomb and image contributions are not calculated separately. Instead, a combined “EDL” contribution to the chemical potential and free energy is calculated by solving the Poisson equation for the electrostatic potential ($\varphi(\mathbf{r})$). The corresponding chemical potential is calculated as

$$\mu_{EDL}^{ex}(\mathbf{r}) = \sum_i q_i \varphi(\mathbf{r})$$

and the EDL contribution to the free energy is calculated as

$$F_{EDL}^{ex} = \sum_i \int_{\Omega} q_i \rho_i(\mathbf{r}) \varphi(\mathbf{r}) d\mathbf{r}$$

which provides an exact solution for first-order electrostatics. The latter expression is now provided in SI on page 15.

c) *What is the explicit expression, particularly the density dependence, for $F^{ex}_{ion\ vDW}$?*

Response: We now provide the expression for $F^{ex}_{ion\ vDW}$ on page 17 of the revised SI: The corresponding component of the excess free energy for two particles separated by a distance D along the z-axis is calculated as

$$F_{ion\ vDW}^{ex} = \int_{-D/2}^0 dz \sum_i \rho_i(z) V_i^{ion\ vDW} \left(|z - \frac{D}{2}| \right) - \int_0^{D/2} dz \sum_i \rho_i(z) V_i^{ion\ vDW} \left(z + \frac{D}{2} \right)$$

d) *What was the size of the computational cell and its boundary conditions? Did the size vary from one calculation to another?*

Response: To address this question the following text was added to the SI on page 18: “The basic system geometry used for all cDFT simulations consists of two rectangular parallelepiped particles separated by a distance D along z-direction, the (x,y) area of the particles is 10 nm × 10 nm (Figure 5a in the main text). The particles are surrounded by the 10 nm³ solvent bath from both sides. Newman boundary conditions has been used.”

e) *What numerical method was used. If it is finite elements, what was the lattice spacing? If pseudo-spectral, what were the basis functions? I realize that it would not be appropriate to give all details but a general description of the method would be helpful.*

Response: To address this question the following text was added to the SI on page 18: "cDFT equations are discretized by a finite difference scheme and solved iteratively using the Gummel method with relaxation. The algebraic multigrid method is applied to efficiently solve the Poisson equation. A novel strategy for calculating excess chemical potentials through fast Fourier transforms is implemented, which reduces computational complexity from $O(N^2)$ to $O(N\log N)$, where N is the number of grid points. Integrals involving the Dirac delta function are evaluated directly by coordinate transformation, which yields more accurate results compared to applying numerical quadrature to an approximated delta function (see ref.²⁴ for details)."

f) Were the calculations performed at constant particle number of chemical potential? If at constant particle number, how was the number chosen in each calculation? If at constant chemical potentials, how were those chosen?

Response: The calculations were performed at constant particle number. The following text has been added to the SI on page 18: "The uniform distribution of all mobile species with their bulk densities in the available solution space of the simulation cell was used as the initial guess. The grid spacing of 0.02 nm, which corresponds to approximately 1/10 of particle diameter, was used."

3. I note that on the last page of the SI, the text reads : "Density profiles are calculated within cDFT via the minimization of the excess free energy..." - this must be a typo as one minimizes the total (not just the excess) free energy functional.

Response: We thank the Reviewer for pointing out the typo. The sentence was corrected accordingly and reads: "Density profiles are calculated within cDFT via the minimization of the total free energy..."

4. How were the disjoining pressures calculated? I suppose it was by varying the separation of the particles and then differentiating the free energy with respect to the distance between faces. Is that correct?

Response: This is correct. To clarify this aspect the following text was added on page 14 of the SI: "Disjoining pressure was calculated by varying separation between the particles with the step of 0.25 nm for separations lower than 3 nm and the step of 0.5 nm for larger separations. Then the resulting free energy per unit area was differentiated with respect to interparticle separation."

5. The large oscillations in the pressure at large separations are quite puzzling, as noted by the authors. It would be helpful if they would show the corresponding density profiles. Do they also show such large oscillations? Can the oscillations be suppressed by turning off one of the terms, e.g. the Coulombic term in the energy?

Response: As we showed through the comparison of the results of full cDFT and primitive models (Fig. 5), the oscillations are suppressed when solvent is treated as a mean-field medium. Effectively, oscillations are suppressed if solvent-related components of the free energy, F_{hs} , which gives rise to interfacial solvent structuring, and ion solvation, F_{solv} , are turned off. We note that oscillations are only observed in the low ionic strength regime, when solvent structuring becomes the most efficient mechanism for balancing the surface charge of the nanoparticles. The oscillations are not observed when counterion concentration is increased (Fig. 5e). The oscillating phenomena are indicative of solution correlations arising and the fact that the isotropic bulk is not achieved at large separations is a cause for some concern. Nevertheless, to the extent that our full cDFT model foreshadows interesting correlated phenomena in a fully interacting system is a point that we want to establish. Future work will be focused on establishing the validity of the explicit hard-sphere solvent and non-electrostatic mean field solvent-ion interactions that have been added to the cDFT framework through self-consistent checks with direct simulation (using Monte Carlo) of the full model. We believe that this will articulate the role of surface characteristics that are typically short-ranged that could have strong coupling to the observed long range nature of the nanoparticle interactions.

We added density profiles of ions and solvent (Figure S10) in the SI, along with some discussions, as additional information for the full cDFT model. See below:

We consider the meaning and implications of the oscillations seen in the full cDFT simulations for separations between 4 and 8 nm. These features are an unexpected prediction that have not been previously reported, nor are they obviously reflected in the experimental observations, though the small magnitude of the predicted barriers ($\sim kT$ for 2 nm particles) and the significant fluctuations in the experimentally derived RDFs may preclude such comparisons in the current data sets. Nonetheless, some discussion of their source and potential implications are in order. First, their absence in the primitive models is not surprising. In contrast to the full cDFT model used in this study, primitive cDFT models that do not treat solvent or ion-solvent interactions explicitly.

Second, the common qualitative feature of full and primitive cDFT models is the dominance of ion correlation forces in interparticle interactions at long- to medium-range separations. However, the differences in ion correlation forces calculated using these two cDFT models further emphasizes the importance of the role of solvent in producing a predictive model of Zn-ion solvation interactions. The peaks and the wells in the disjoining pressure calculated in the full cDFT model correspond to the energetic cost and gain for accommodating a fraction or a whole number of structured solution layers between the surfaces, respectively (Fig. S10). This aspect of the model needs to be further validated with direct simulations (via Monte Carlo) of the full cDFT model. Moreover, the role of the discreteness of charge distribution on the interacting surfaces that will increase the ion correlation forces needs to be further explored in connection

to orientation specificity.^{1,2} Because these oscillations are directly related to the solution structuring predicted in the full cDFT model (see Fig. S10), our results suggest that such oscillations should be observed in experiments that probe solution-solid interfaces between two surfaces^{2,3} in hopeful *qualitative* agreement with the full cDFT model. If they cannot be observed, then the possibility they are an artifact of the fully parameterized cDFT method itself needs to be explored.

Figure S10. Density profiles of ions and solvent confined between oppositely charged (0001)-(000-1) faces calculated using a full cDFT model for 1 mM zinc acetate dehydrate solution in methanol. The distances between the surfaces are shown in the insets.

6. How was the interaction potential in figure S9 calculated?

Response: The following text was added on page 15 of the SI to address this question: “Calculations to implement the appreciable curvature of the particle were performed based on the surface element integration scheme⁴ which provides a higher accuracy than the Derjaguin approximation. In this method interactions between surface elements facing each other are calculated and integrated over the equidistant shells over the surfaces facing each other, surfaces facing in the opposite directions and the corresponding cross terms are evaluated based on the separation dependence of free energy and potential between flat surfaces.”

References

- 1 Sushko, M. L. *et al.* The Role of Correlation and Solvation in Ion Interactions with B-DNA. *Biophys J.* **110**, 315-326, (2016).
- 2 Zhang, X. *et al.* Accessing crystal–crystal interaction forces with oriented nanocrystal atomic force microscopy probes. *Nature Protocols* **13**, 2005-2030, (2018).
- 3 Zhang, X. *et al.* Direction-specific van der Waals attraction between rutile TiO² nanocrystals. *Science* **356**, 434-437, (2017).
- 4 Bhattacharjee, S., Elimelech, M. & Borkovec, M. DLVO interaction between colloidal particles: Beyond Derjaguin's approximation. *Croat Chem Acta* **71**, 883-903 (1998).

REVIEWERS' COMMENTS:

Reviewer #1 (Remarks to the Author):

The manuscript have been improved. Since it is of great significance yet challenging in evaluating the driving force of OA, it would be better to draw the conclusion relatively cautiously. In particular, the separation distance of attractive interaction and torques need to be defined more carefully. Note that a distance can be defined from the surface of one particle to that of another. If this is the case, however, from Figure 3a-c, it seems many particles of a distance approaching 5-10 nm have no obvious contact with each other based on experiments. From Figure 5, it looks like the distance was defined from the center to center. In this case, for the >5 nm particles of similar 10-nm separation distances, it actually reflects a small surface-to-surface distance value. It has been generally accepted that the number of surface atom significant change within tiny nanocrystals, such as 63% surface atom for 3-nm particle and 45% for 5-nm particle. In such small surface-to-surface distance, it is hard to exclude the influence of particle size and surface atom number which might impact the contact area, the interaction force strength or even the type, and subsequently the accurate action distance analysis. As the merit of this work is to discuss the type and distance of attractive interaction based on both experiment and modelling, it seems the particle size need to be carefully declared and taken into account when drawing or discussing the conclusion. Finally, it is impressive for conducting a solid calculation that well fit the experimental data and for a very detailed analysis and discussion as well. In addition, I suggest the author add some review on progress in this aspect, driving force or attractive distance of OA in Introduction Part.

Reviewer #2 (Remarks to the Author):

In this revised manuscript, authors clarified all the comments I mentioned in previous review report. Especially my last question, authors give a clear statement on their key conclusion that the dipole-dipole interaction is the dominant effect in the orientation attachment of ZnO nanoparticle growth. Now the manuscript looks better, and I have no objection to accept it for publication.

Reviewer #3 (Remarks to the Author):

I have read the revised manuscript, the reviews and the authors' responses to the reviews. In my previous review I praised the experimental work and asked for several points of clarification, mostly concerning the theoretical support. I find that the necessary clarification has been given and that the manuscript is suitable for publication.

Reviewer #1 (Remarks to the Author):

The manuscript have been improved. Since it is of great significance yet challenging in evaluating the driving force of OA, it would be better to draw the conclusion relatively cautiously. In particular, the separation distance of attractive interaction and torques need to be defined more carefully. Note that a distance can be defined from the surface of one particle to that of another. If this is the case, however, from Figure 3a-c, it seems many particles of a distance approaching 5-10 nm have no obvious contact with each other based on experiments. From Figure 5, it looks like the distance was defined from the center to center. In this case, for the >5 nm particles of similar 10-nm separation distances, it actually reflects a small surface-to-surface distance value. It has been generally accepted that the number of surface atom significant change within tiny nanocrystals, such as 63% surface atom for 3-nm particle and 45% for 5-nm particle. In such small surface-to-surface distance, it is hard to exclude the influence of particle size and surface atom number which might impact the contact area, the interaction force strength or even the type, and subsequently the accurate action distance analysis. As the merit of this work is to discuss the type and distance of attractive interaction based on both experiment and modelling, it seems the particle size need to be carefully declared and taken into account when drawing or discussing the conclusion. Finally, it is impressive for conducting a solid calculation that well fit the experimental data and for a very detailed analysis and discussion as well. In addition, I suggest the author add some review on progress in this aspect, driving force or attractive distance of OA in Introduction Part.

Response: We thank Reviewer #1 for the feedback.

- In fact, the analysis of the DFT data (Figure 5) is also in terms of the edge-to-edge particle separation (h), as described in Figure 5a, and can hence be directly compared to the experimental data (Figure 3). To ensure that the separations used throughout the paper are measured from edge-to-edge, rather than center-to-center, in every location where the distinction is relevant and potentially ambiguous, we have added “edge-to-edge”. Instances occur on pages 4, 6, 8, 12, 13, 15, 17, 18, and 22 and Figure captions 3, 5 and 6.
- To further clarify the particle size in the manuscript, we have added a phrase to the text on page 18 to read:

“The findings reported above show that ZnO nanoparticles of approximately 5 nm diameter are attracted and experience an aligning torque...”

The particle diameter is already specified in all other instances.

- As requested, we have added a section on the driving forces for OA in the Introduction on page 2, which now reads:

“In the case of iron oxyhydroxide, two nanoparticles approach each other and rotate continuously until their lattices are perfectly aligned. At that moment, the particles jump to contact and fuse into a single crystal.⁹ Analysis of the particle motion showed that

Coulomb forces alone could account for the jump to contact with an effective difference of only one net unit charge on the surface of the particles, but a detailed analysis of other potential forces was not carried out. By comparison, ligand-coated gold nanoparticles experience a short-range barrier due to steric repulsion before undergoing attachment.²⁹ In the case of gold nanorods, the potential energy as a function of angle and distance was derived for that system, the underlying force components (e.g., van der Waals, dipole-dipole, etc.) were not examined. Thus, despite direct evidence of OA, the nature and scaling of the underlying forces is not understood and hence a comprehensive understanding of the mechanism remains lacking.”

Reviewer #2 (Remarks to the Author):

In this revised manuscript, authors clarified all the comments I mentioned in previous review report. Especially my last question, authors give a clear statement on their key conclusion that the dipole-dipole interaction is the dominant effect in the orientation attachment of ZnO nanoparticle growth. Now the manuscript looks better, and I have no objection to accept it for publication.

Response: We thank Reviewer #2 for their assessment and initial feedback.

Reviewer #3 (Remarks to the Author):

I have read the revised manuscript, the reviews and the authors' responses to the reviews. In my previous review I praised the experimental work and asked for several points of clarification, mostly concerning the theoretical support. I find that the necessary clarification has been given and that the manuscript is suitable for publication.

Response: We thank Reviewer #3 for their assessment and initial feedback.